# Immunophysiological State of Dogs According to the Immunoregulatory Index of Their Blood and Spleens

**DOI:** 10.3390/ani14050706

**Published:** 2024-02-23

**Authors:** Oksana Dunaievska, Ihor Sokulskyi, Mykola Radzykhovskii, Bogdan Gutyj, Olga Dyshkant, Zoriana Khomenko, Viktor Brygadyrenko

**Affiliations:** 1Department of Normal and Pathological Morphology, Hygiene and Expertise, Faculty of Veterinary Medicine, Polissya National University, Stary Boulevard Str. 7, 10002 Zhytomyr, Ukraine; guta_1985@ukr.net (O.D.); sokulskiy_1979@ukr.net (I.S.); zorianakhomenko@ukr.net (Z.K.); 2Department of Epizootology, Microbiology and Virology, Faculty of Veterinary Medicine, National University of Life and Environmental Sciences of Ukraine, Heroiv Oborony Str. 15, 03041 Kyiv, Ukraine; nickvet@ukr.net (M.R.); olga@ukr.net (O.D.); 3Department of Hygiene, Sanitation and General Veterinary Prevention, Faculty of Public Development and Health, Stepan Gzhytskyi National University of Veterinary Medicine and Biotechnologies Lviv, Pekarska Str. 50, 79010 Lviv, Ukraine; 4Department of Zoology and Ecology, Oles Honchar Dnipro National University, Gagarin Av. 72, 49010 Dnipro, Ukraine; brigad@ua.fm

**Keywords:** morphology, immunohistochemical studies, subpopulations of T-lymphocytes, histological preparation, immunoregulatory index

## Abstract

**Simple Summary:**

Immunodiagnosis is essential for studying possible changes in the immune system, which, together with the nervous and endocrine systems, constitutes the only necessary regulatory system of the body. Among such studies, the state of the cellular link of immunity is critical. These studies play an essential role in preventing diseases and increasing the effectiveness of diagnostics. The immunoregulatory index, which allows us to assess the immunophysiological status of the direction of adaptive processes and the risk of autoimmune damage to cellular structures, is of critical importance in assessing the state of the immune system. The immunoregulatory index is sensitive to various effects and diseases of the body: liver disease, obesity, mononucleosis, and radionuclide contamination of an area. Scientific studies demonstrate that this index depends on the breed characteristics of animals, has seasonal fluctuations, and is proposed for determination in the selection process for the genetic improvement of already existing breeds. In current conditions, the use of service dogs to search for explosives, people under rubble, and narcotic substances is increasing. It is essential to observe the state of animals’ health because the quality of their official duties will depend on it. In this study, the immunoregulatory index was evaluated in dogs, which allows us to objectively assess the immune status of their body and prevent the development of a secondary immunodeficiency state.

**Abstract:**

In this study, the immunological characteristics of a dog’s body were established, allowing for a quick reaction to any changes in the immune status and the development of an immunodeficiency state. The immunoregulatory blood index was determined to indicate the ratio of T-helpers and T-suppressors. The immunoregulatory index of the spleen was determined as the ratio of CD4+ cells to CD8+ cells in the field of view of a microscope (eyepiece 10, objective 40) after obtaining histological preparations according to generally accepted methods. It was found that the number of T-helpers decreased by 0.13 × 10^12^/L, while the number of T-suppressors increased non-significantly by 0.01 × 10^12^/L after intensive exercise during tasks. The immunoregulatory blood index of dogs was 2.1 ± 0.1 and 1.7 ± 0.13 before and after intensive exercise, respectively. Lymphocytes with markers CD4+ and CD8+ were located almost all in the white pulp; in the red pulp, they were found alone, and their share was 3.4% and 1.9%, respectively. Lymphocytes with CD4+ markers in the spleen’s white pulp were mainly concentrated in lymphoid nodules (60.7%), of which 20.1% were focused on the marginal zone, and slightly less in the light center (19.4%) and the periarterial zone (18.1%). Lymphocytes with CD8+ markers in the spleen’s white pulp were also mainly concentrated in lymphoid nodules, but their number was 8.1% higher (68.8%). The immunoregulatory index of the spleen is 1.9. These findings emphasize the need for the assessment of the immunoregulatory index in service dogs to prevent the development of secondary immunodeficiency and allow them to properly perform their official duties.

## 1. Introduction

Immunodiagnosis is essential for identifying changes in the immune system [1,2]. The study of the cellular link of immunity is essential not only for the study of pathological conditions of the body but also in the case of increased physical exertion. In athletes, immunoreactivity decreases after a disease that cannot be recovered from due to increased physical exertion [3]. Such research made it possible to establish the leading place for changes in the immune system as a mechanism for the development of type 2 diabetes [4]. Thanks to immunological studies, it has been proven that the probability of pneumonia in patients who are oncohematological is determined by the state of the immune system [5]. Studying the links of immunity made it possible to prove the feasibility of using cryopreserved fetal liver cells for the treatment of breast cancer [6]. Such studies show the immunocorrective effect of drugs, particularly lyophilized and cryopreserved leukoconcentrate of cord blood in atypical dermatitis [7]. Quantitative and functional immunological disorders play a significant role in the pathogenesis of many diseases, particularly cardiovascular, respiratory, obstetric, and gynecological diseases [8,9].

The immunoregulatory index (IRI) plays a leading role in assessing the state of the immune system [10]. IRI makes it possible to assess the immunophysiological status [11]. It also characterizes the direction of adaptive processes. An increase in IRI above >2.5 affects the risk of autoimmune damage to cellular structures and corresponds to the hyperactivity of the immune response in patients with type 2 diabetes and non-alcoholic fatty liver disease [4]. A significant decrease in IRI indicates significant violations of immune protection and a more severe course of the pathological process [12]. The balance between regulatory (T-helper CD4+) and effector (T-suppressor CD8+) cells affects the pathogenesis of various immune-dependent liver diseases [13]. Scientific studies have established that IRI decreases in residents living in a territory contaminated with radionuclides [14], in infectious mononucleosis [15], in athletes [16], and in acne [1]. IRI decreases in the case of atypical mucous membrane changes in cervical intraepithelial neoplasia. In this case, immunological studies can detect the transitional form of the disease and prevent the development of cervical cancer, increasing the effectiveness of diagnostics [8]. In veterinary medicine and animal husbandry, much attention is paid to modern methods of researching the state of animal health [17,18]. Immunological studies, particularly the establishment of IRI, play a significant role [19,20,21]. Scientific studies demonstrate that this index depends on the breed characteristics of animals and is proposed to be determined in the selection process in the conditions of industrial cattle breeding for the genetic improvement of already existing breeds [22,23]. Local inflammatory processes, such as catarrhal gingivitis in dogs, lead to changes not only in the place of pathology but also have a systemic effect. This is manifested in the development of immunological insufficiency and reduction in the IRI [24]. The IRI in dogs has seasonal fluctuations: the indicator has the highest value in winter and the lowest in spring. It is interesting to observe the value of the IRI in the same animals with the same diagnosis but in the presence of another diagnosed disease. Thus, in rats with acute periodontitis, its value increases compared to intact animals; in rats with hypothyroidism, it decreases, and in animals with simultaneous manifestation of acute periodontitis and hypothyroidism, the IRI significantly decreases [25]. The IRI is reduced in patients with non-alcoholic fatty liver disease and chronic hepatitis C. However, in patients with non-alcoholic fatty liver disease, the content of CD8 lymphocytes does not change [26]. The authors’ research also revealed a relationship between the absolute number of leukocytes, lymphocytes, phagocytic activity, and the IRI [27]. Recognizing the importance of assessing the immune status, it is difficult to develop methods based on the results of which it is possible to determine this status in an individual or group and then extrapolate it to the population [11].

In current conditions, the use of service dogs to search for explosives, people under rubble, and narcotic substances is increasing. Today, dogs are used by Ukraine’s criminal enforcement service. They not only perform search and guard functions but also psychologically influence persons held in places deprived of liberty [28,29].

Specialists in the canine service have the critical task of selecting and training such dogs. Monitoring the health of the animals is an integral part of this work because the performance of their official duties will depend on it.

This work aimed to establish the immunological characteristics of a dog’s body, which will allow it to quickly react to a change in the immune status and the development of an immunodeficiency state.

## 2. Materials and Methods

### 2.1. Experimental Condition, Research Ethics and Procedures

All experimental studies were carried out according to modern methodological approaches and in compliance with the relevant requirements and standards; in particular, they meet the requirements of SSTC ISO/IEC 17025:2017 [30]. During the research, the basic rules of good laboratory practice GLP (1981) as well as the provisions of the “General Ethical Principles of Animal Experiments” adopted by the First National Congress of Bioethics were followed. The entire experimental part of this study was carried out following the requirements of the international principles of the “European Convention for the Protection of Vertebrate Animals Used in Experiments and for Other Scientific Purposes”, “Rules for Conducting Work Using Experimental Animals”, approved by Order of the Ministry of Health No. 281 dated 1 November 2000 “On measures to further improve organizational forms of work using experimental animals” and the corresponding Law of Ukraine “On the Protection of Animals from Cruelty” (No. 3447-IV dated 21 February 2006, Kyiv) [31].

During 2022–2023, the condition of the cellular immunity of the body of dogs, which were selected according to the principle of analogs, was studied in the conditions of the canine center of the Ministry of Health of the Zhytomyr region. Two groups of 5–6-year-old service dogs were formed, which were examined according to the veterinary surveillance program in March. The first (control) group was examined for T-helper and T-suppressor levels in the blood before the participation of dogs in ensuring the official activities of the National Police of Ukraine (*n* = 56). A re-examination was performed 24 h after the dogs were returned to their housing (*n* = 46); thus, this was the experimental group. T-lymphocyte subpopulations were determined in the reaction of rosette formation with ram erythrocytes (JSC Pharmstandard Biolek, Kharkiv, Ukraine).

Spleens were removed from animals that died (*n* = 6). Anatomical dissection was performed using the method of complete evisceration. Anatomical preparation of the spleen of the studied animals was quickly performed (within 30–40 min after the animal’s death). For histological studies, pieces of 1 cm^3^ were taken from the visceral surface of the spleen in the portal area, the central part of the median plane, and in the middle of the parietal surface in the capsule area (at least five pieces were taken from each spleen). Spleens were fixed in a 10–12% chilled aqueous solution of neutral formalin for histological studies. The fixed material for further research was washed with running water for 24–48 h. Then, they were dehydrated in ethyl alcohol of increasing strength—40°, 70°, 96°, 100°—and poured into paraffin according to the scheme proposed in the manual [32]. Paraffin sections were made on a sled microtome MC-2. The thickness of the sections did not exceed 4–10 μm. For the morphological characterization of the spleens, tissue sections were stained with Ehrlich’s hematoxylin (Diapath, Martinengo, Italy, 2020) and eosin (Leica Geosystems, Garching bei München, Germany, 2020). Lymphocyte subpopulations (CD4+, CD8+) were detected using mouse monoclonal antibodies (DACO, Glostrup, Denmark). Visualization was performed using the DAKO EnVision FLEX+ detection system (DAKO, Glostrup, Denmark) with additional staining with Mayer’s hematoxylin (DAKO, Glostrup, Denmark) for 1–3 min and transfer to Eukitt medium (Freiburg, Germany).

The immunoregulatory index (IRI) of the spleens was determined as the ratio of the number of CD4+ cells to CD8+ cells in the field of view of a light microscope (eyepiece ×10, objective ×40) at a magnification of ×400. Blood IRI was determined to indicate the ratio of T-helpers and T-suppressors.

Photomicrographs of histological preparations were carried out using a CAM V-200 video camera (Inter Med, Shenzhen, China, 2017), mounted in a Misros MC-50 microscope (Sankt Veit an der Glan, Austria, 2012), and connected to a personal computer.

The International Veterinary Histological Nomenclature [33] and the International Veterinary Anatomical Nomenclature [34] were followed when describing the histological structures of the spleen established during the research.

### 2.2. Statistical Analyses of Data

The statistical analysis of the obtained data provided for determining the nature of the distribution was carried out according to the Shapiro–Wilk test (*p* ≤ 0.05). After establishing the normal distribution, Tukey’s test (*p* ≤ 0.05) was applied in the SPSS statistical package. The correlation between indicators was analyzed according to Spearman’s correlation (*p* ≤ 0.05). The mean value (mean) and the standard deviation error (SDE) were determined.

## 3. Results

The indicators of the cellular immunity of dogs were determined by the indicators of T-helpers and T-suppressors in the blood and the calculation of the immunoregulatory index. A clinical examination of the animals was carried out beforehand. The dogs were determined to be healthy and fit for service based on the results of a general clinical blood test (number of erythrocytes, leukocytes, lymphocytes, monocytes, and platelets). The results of our study are shown in Table 1.

Thus, the immunoregulatory index significantly decreased after using dogs in particular works by 0.4 (during physical exertion (combined running and walking) and psycho-emotional) (Table 1). Such results must be considered to calculate the time for the recovery of service dogs to successfully carry out the service on which people’s lives often depend. The IRI decreased. Accordingly, the number of T-helpers and T-suppressors changed. The number of T-helpers decreased by 0.13 × 10^12^/L, while the number of T-suppressors increased insignificantly by 0.01 × 10^12^/L.

The spleens of sexually mature dogs are fully formed. Their histoarchitectonics consist of the support-contractile apparatus and the pulp, characteristic of the spleens’ vertebrates. The support-contractile apparatus includes a capsule and trabeculae of various types. The pulp is divided into white and red. The white pulp was divided into lymphoid nodules and periarterial lymphoid sheaths in the histological examinations. Lymphoid nodules had a heterogeneous structure; separate zones were distinguished in their structure: a light center, an artery with a periarterial zone, a mantle zone, and a marginal zone. Our immunohistochemical studies established that subpopulations of lymphocytes with CD4+ and CD8+ markers were unevenly distributed in the spleen pulp of dogs. The vast majority of such lymphocytes were in the white pulp. In the red pulp, lymphocytes with CD4+ spleen markers were found singly or in small groups; instead, they were scattered, often near the connecting trabeculae, and such lymphocytes were mainly polygonal in shape (Figure 1a). Certain features of the localization of such lymphocytes are inherent in lymphoid nodules. Yes, they had the largest concentration in the light centers of lymphoid nodules. Cell concentrates of 2–3 lymphocytes with CD8+ markers were visible here. Their concentrations were significant in the periarterial zone of nodules; cells with CD4+ and CD8+ markers were closely adjacent to the outer shell of vessels (Figure 1b). T-cells with CD4+ markers were observed in small numbers in the marginal zone of lymphoid nodules. In the mantle zone, there were significantly fewer cells with CD4+ markers than in the light center of the lymphoid nodule (Figure 2a). There were slightly less lymphocytes with CD8+ markers in this zone than cells with CD4+ markers. In the periarterial lymphoid sheaths of the spleen’s white pulp, CD4+ and CD8+ lymphocytes were found over the entire area without being concentrated in any particular part (Figure 2b).

Lymphocytes with CD8+ markers were also detected in the red pulp, but these were individual cells that were randomly located, rarely grouped by 2–3 cells.

According to the results of morphometric immunohistochemical studies of a dog’s spleen, the number of CD4+ lymphocytes in the periarterial lymphoid sheaths of the spleen was 36.3% of the total population of the pulp, and the rest (60.7%) was located in the lymphoid nodules and only 3.4% in the red pulp (Figure 3).

In the structure of lymphoid nodules, such lymphocytes are unevenly distributed: the most significant number was detected in the marginal zone of the nodules, the periarterial zone, and light centers (Figure 4a,b). Thus, in the marginal zone of nodules, the number of CD4+ lymphocytes was 23.2 ± 6.1 units per unit area, respectively; in the light center of lymphoid nodules, their number was 22.4 ± 5.2 units per unit area, respectively; and in the periarterial zone, their number was 20.9 ± 2.4 units per unit area, respectively. In sum, they amounted to 20.1%, 19.4%, and 18.1%, respectively. In the mantle zone of lymphoid nodules, the number of CD4+ lymphocytes was the lowest and amounted to 3.6 ± 0.6 units per unit area, respectively (Figure 5).

The number of CD8+ lymphocytes was lower than CD4+ lymphocytes in the periarterial lymphoid sheaths of the spleen by almost 1.2 times—29.2% and 36.3%, respectively. In particular, 17.8 ± 3.6 cells were counted in the periarterial lymphoid sheaths with CD8+ markers (29.2% of the total population in the white pulp). In lymphoid nodules of the spleen, their number prevailed by 2.3 times, compared to the periarterial lymphoid sheaths, and was 41.2 ± 10.4 units per unit area, respectively. Moreover, they were often densely located and occupied a significant part of the field of vision (Figure 5).

Cytomorphometric studies established that the number of lymphoid nodules in this population’s mantle zone was much smaller than the light center of lymphoid nodules, where they were found 4.8 times more—15.4 ± 4.7 units per unit area and 3.2 ± 0.4 units per unit area, respectively (Figure 5).

In the periarterial zone of lymphoid nodules, the number of CD8+ lymphocytes was 12.2 ± 2.4. units per unit area. A large number of lymphocytes with CD4+ markers compared to the number of CD8+ lymphocytes of the splenic pulp led to a high IRI value. The immunoregulatory index of the white pulp was equal to 1.9 ± 0.11.

## 4. Discussion

One of the urgent tasks of laboratory medicine is the study of immunohistochemical markers for pathological conditions of various geneses [35]. Long-term experimental studies show that the course of general syndromes of dependence on the changes in the body’s functioning is accompanied by the development of universal pathological processes, such as endogenous intoxication and impaired immunological reactivity of the body. The expressiveness of the manifestations of these processes indicates their leading importance in forming central metabolic disorders, which can be used in laboratory diagnostics to assess the degree of severity and predict the course of diseases in humans and animals [36]. The basis of the multicomponent blood system’s function is the basic principle of a living system—stability in the face of its constant dynamic variability [37]. The multifunctional nature of an animal organism’s organization creates conditions for the functioning of energy information flows between blood cells, tissues, and organs, determining the intensity and direction of regenerative and reparative processes in the body [38,39].

It is necessary to note that the blood system is not only an automated complex structure but also a complex of functional components that are included in the system and fall out of it as it functions, coming from tissues and organs. The level of available activity in the blood can increase abruptly when physiological functions deviate from the optimal level of metabolism [40]. Thus, maintaining the health of animals is an essential task for owners and farms; the determination of immunity indicators, particularly lymphocytes with markers CD4+ and CD8+, is of increasing importance [41]. Scientific studies have established that for the prevention of immunodeficiency, the addition of a casein–phosphor–peptide–selenium complex to the primary diet of dogs contributed to an increase in the number of such lymphocytes in the peripheral blood [42].

Our research has studied the state of cellular immunity in 5–6-year-old service dogs after physical and psycho-emotional stress. Other scientists established the dependence of the immunoregulatory index in dogs depending on the season [27]. The results of our research were obtained in the spring; they do not significantly differ from the research of other scientists. So, the IRI was equal to 2.1 ± 0.1 and 2.33, respectively. Such a discrepancy in research results can be explained by age—in Broshkov M. M. (2014) [11], the age group ranged from 1 to 6 years, while the age of the dogs in our control group was limited and ranged from 5 to 6 years. At the same time, in the summer, the IRI increased concerning the value in spring, and the month of study was not specified in this work. For example, the IRI should be higher in May than in March. The control group of dogs was subjected to a significant load of various natures. Broshkov M. M. (2014) [11] determined the immunoregulatory index in 2-month-old puppies, which had an average value of 3.7; the same author indicates the physiological limits of the IRI as 2–4. Since dogs belonging to service breeds are not involved in performing tasks at such an age, we have not studied such information. Accordingly, the IRI has not only seasonal but also age-related fluctuations. We drew attention to the study of the immunoregulated index in dogs of a similar age category in a different paper [24]. However, the IRI values differed—1.22 and 1.9, respectively. The breed type attracted attention in the following context: we studied dogs belonging to service breeds, while the authors noted purebred pets. Most likely, the conditions of maintenance and feeding were affected. In the same author’s study, attention was drawn to the IRI, which decreased in the presence of chronic catarrhal gingivitis. In the dogs of our research group, the IRI decreased due to the loads, which may lead to the development of a secondary immunodeficiency state when the loads increase. In their research, Kyrychenko et al., 2023 showed that estrus affects the content of CD4+ T cells in the blood of female Labrador dogs. Moreover, the content of T-helpers depended on the phase of the sexual cycle: on the first day, their share was 43.3 ± 2.07%; on the fifth—50.0 ± 1.78%; on the tenth—48.6 ± 1,03%; on the twentieth—51.3%; and on the twenty-fifth—47.3% [43]. Research by Verde M.T. et al., 2023, established that the IRI value positively correlates with the degree of severity of atopic dermatitis in dogs (3.73 ± 1.8 and 2.98 ± 1.74—control and experiment, respectively), thereby confirming the importance of determining T-lymphocyte subpopulations during dispensation and conducting a thorough examination of animals, especially after completing tasks [44].

Our histochemical studies established that in the mantle zone, there was a lower number of lymphocytes with markers CD4+ and CD8+ than in other structures of lymphoid nodules of the spleen’s white pulp (bright center, marginal zone, periarterial zone). This is due to the peculiarities of the lymphoid nodules’ structure because the mantle zone surrounds the light center and the periarterial zone with a thin layer of one, sometimes two, densely arranged row(s) of cells. Most often, lymphocytes with markers CD4+ and CD8+ were detected around the periarterial lymphoid sheaths, which is explained by the high filtration property of such a structure. A significant proportion of lymphocytes with CD4+ and CD8+ markers around the central artery of lymphoid nodules in the periarterial zone confirms our previous hypothesis. A significant number of lymphocytes with markers CD4+ and CD8+ were concentrated in the marginal zone because it is in this part of the spleen’s white pulp where the cooperative interaction of immune cells takes place. Such an uneven density of CD4+ and CD8+ lymphocytes in the spleen’s white pulp is possibly due to the formation of immune response stages in lymphoid nodules under antigenic stimulation.

## 5. Conclusions

Our research has shown that the immunophysiological state of the body changes the physical work and psycho-emotional stress of animals. Thus, the immunoregulatory blood index of dogs after intensive exercise significantly decreased from 2.1 ± 0.1 to 1.7 ± 0.13. This change mainly occurred due to a decrease in T-helpers by 13.4% and a slight change in the number of T-suppressors (+2.2%). It should be noted that the recovery of hematological indicators did not fully occur within 24 h. We also studied the peculiarities of the location of lymphocytes with CD4+ and CD8+ markers in the spleen’s pulp. Almost all of them were located in the white pulp; in the red pulp, their share was only 3.4% and 1.9%, respectively, with regard to the total number of lymphocytes with CD4+ and CD8+ markers. Furthermore, certain features of their localization were established: lymphocytes with both CD4+ and CD8+ markers in the spleen’s white pulp were mainly concentrated in lymphoid nodules (60.7% and 68.8%); in the lymphoid nodules themselves, the populations of these lymphocytes were located unevenly; in the periarterial lymphoid sheaths, the white pulp was found in significant quantities (36.3% and 29.2%, respectively). Finally, it was calculated that the immunoregulatory index of the spleen is equal to 1.9 ± 0.11.

## Figures and Tables

**Figure 1 animals-14-00706-f001:**
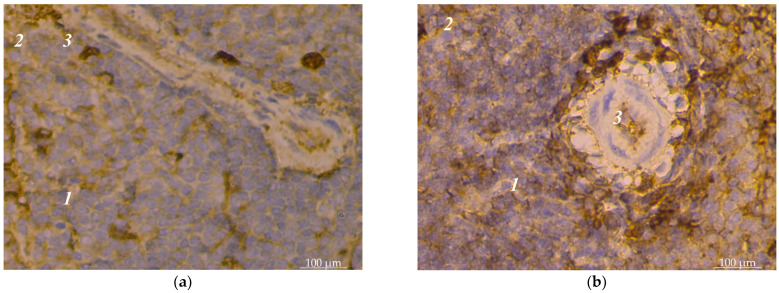
Dog spleen pulp: (**a**) single location of lymphocytes with CD4+ markers: 1—white pulp; 2—red pulp; 3—trabeculae; (**b**) location of lymphocytes with CD4+ markers close to the adventitia of vessels: 1—white pulp; 2—red pulp; 3—artery of the periarterial zone of the lymphoid nodule. Hematoxylin with additional staining with Mayer’s hematoxylin.

**Figure 2 animals-14-00706-f002:**
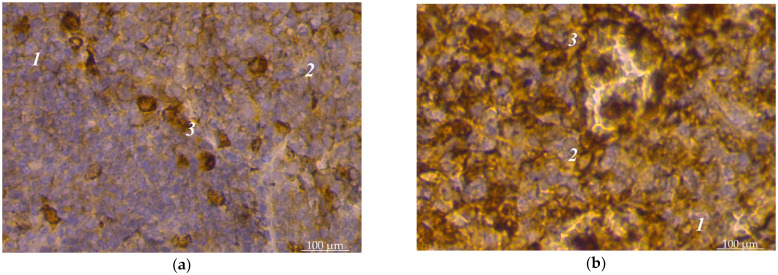
Spleen pulp of dogs: (**a**) location of lymphocytes with CD4+ markers in the mantle zone: 1—fragment of lymphoid nodule of white pulp; 2—bright center; 3—mantle zone; (**b**) location of lymphocytes with CD8+ markers in the periarterial lymphoid sheath: 1—white pulp; 2—periarterial lymphoid sheath; 3—arteries of the periarterial lymphoid sheath. Hematoxylin with additional staining with Mayer’s hematoxylin. ×280.

**Figure 3 animals-14-00706-f003:**
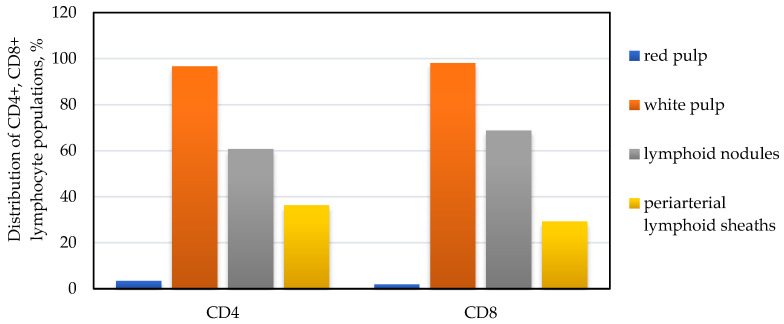
Distribution of CD4+ and CD8+ lymphocyte populations in the spleen pulp of dogs.

**Figure 4 animals-14-00706-f004:**
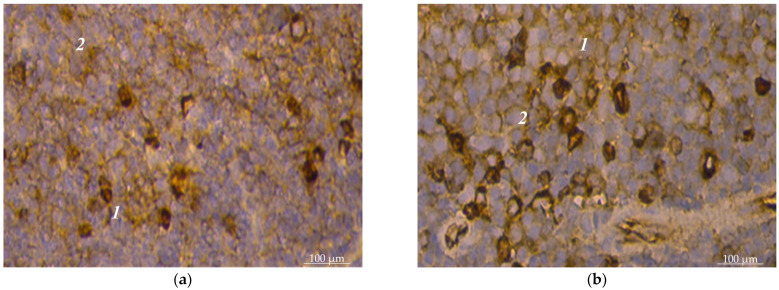
Spleen pulp of dogs: (**a**) location of lymphocytes with CD8+ markers in the bright center of a lymphoid nodule of the white pulp: 1—a fragment of a lymphoid nodule of the white pulp; 2—bright center; (**b**) location of lymphocytes with CD4+ markers in the marginal zone of the lymphoid nodule of the white pulp: 1—lymphoid nodule; 2—marginal zone. Hematoxylin with additional staining with Mayer’s hematoxylin. ×280.

**Figure 5 animals-14-00706-f005:**
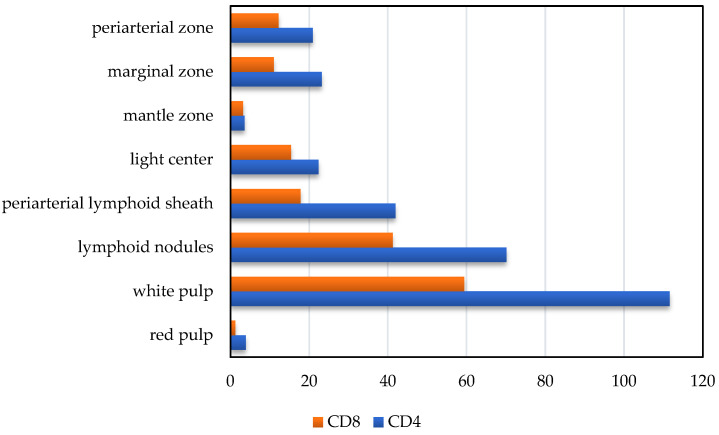
The density of lymphocytes with CD4+ and CD8+ markers in the spleen pulp of dogs.

**Table 1 animals-14-00706-t001:** Indicators of cellular immunity of the dogs’ blood.

Indicator	Control Group of Dogs (*n* = 56)	Experimental Group of Dogs (*n* = 46)
T-helpers, 10^12^/L	0.97 ± 0.08 ^a^	0.84 ± 0.10 ^a^
T-suppressors, 10^12^/L	0.46 ± 0.05 ^a^	0.47 ± 0.09 ^a^
IRI	2.1 ± 0.1 ^b^	1.7 ± 0.13 ^a^

Note: letters indicate significant differences between the subgroups within one line (*p* < 0.05) according to Tukey’s test.

## Data Availability

The data that support the findings of this study are available upon request from the corresponding author. The data are not publicly available due to privacy or ethical restrictions.

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
