# Peer review of "Immunophysiological State of Dogs According to the Immunoregulatory Index of Their Blood and Spleens"

_animals, 2024, doi:10.3390/ani14050706_

Round 1
Reviewer 1 Report
Comments and Suggestions for Authors
The authors presented the results of a study aimed to establish the immunological characteristics of the dog's body, by determining the immunoregulatory index (IRI, the ratio of T-helper and T-suppressor cells) to assess the immunophysiological status in service dogs of the National Police of Ukraine. To carry out this, 56 dogs were examined for T-helper and T-suppressor cell levels in the blood before the participation of dogs in official activities and re-examined 24 hours after the dogs were returned to their housing (n = 46). It was found that the number of T-helper and T-suppressor cells changed, observing that the number of T-helper cells decreased by 0.13 x 1012/L, while the number of T-suppressor cells increased by 0.01·1012/L, a value not statistically significant. However, it was determined that after intensive exercise, the immunoregulatory index significantly was reduced to 1.7. In addition in spleens that were removed from 6 dogs that died it, was determined that the immunoregulatory index of the white pulp was 1.9. The authors conclude that these findings emphasize the need to assess the immunoregulatory index in service dogs to prevent changes in the development of a secondary immunodeficiency state.
The research and its outcomes are of interest to researchers dealing with determining the immunophysiological status of service dogs. Especially those experts who are interested in immunodeficiency diseases under extreme physical and psycho-emotional conditions. The sample size is considered adequate, laboratory analyses are described accurately, and the statistical analysis is correct.
Minor details to be taken care of are as follows:
Lines 36-37. Use “the immunoregulatory blood index of dogs was 2.1 ± 0.1 and 1.7 ± 0.13 before and after intensive exercise, respectively.”
Line 175. Define ESR, first-time use
Lines 181-182. It is stated that “the immunoregulatory index significantly decreased after using dogs in particular works by 0.4”. Authors should define those particular Works.
Line 210. Figure 1. Check and change colors in numbers placed on photomicrograph images. Numbers in black are really difficult to see.
Line 214. Figure 2. Check and change colors in numbers placed on photomycrophag images.
Line 230. Check and correct “units. on the mind unit sq.,” what does it mean?
Line 231. Like wise, “pcs. on the mind unit sq.,”
Line 232. “pcs. on the mind unit sq.,”
Line 234. “units. on the mind unit sq.,”
Line 235. Figure 4. Check and change colors in numbers placed on photomicrograph images.
Line 246. “markers. on the mind unit sq”
Line 248. “pcs. on the mind unit sq.,”
Lines 253 and 254. “pcs. on the mind unit sq.,”
Line 256. “on the mind unit sq.”
Lines 262-265. Rephrase sentence “The search for histochemical studies of markers in the norm and pathologies of organ systems is an urgent task of modern laboratory medicine”
Line 332. Conclusions should be rephrased. Most of the paragraphs in lines 335-348 are a repetition of the findings in the spleen immunohistochemical study described in the results section.
Comments on the Quality of English Language
Minor editing of English language required
Author Response
Thank you very much for the reviewer's report
Lines 36-37. added: the immunoregulatory blood index of dogs was 2.1 ± 0.1 and 1.7 ± 0.13 before and after intensive exercise, respectively.
Line 175. Define ESR, first-time use – deleted
Lines 181-182. It is stated that “the immunoregulatory index significantly decreased after using dogs in particular works by 0.4”. Authors should define those particular Works. - During physical exertion (combined: running and walking) and psycho-emotional
Line 210. Figure 1. Check and change colors in numbers placed on photomicrograph images. Numbers in black are really difficult to see. – corrected
Line 214. Figure 2. Check and change colors in numbers placed on photomycrophag images. – corrected
Line 230. Check and correct “units. on the mind unit sq.,” what does it mean? – corrected (units per unit area respectively)
Line 231. Like wise, “pcs. on the mind unit sq.,” – corrected (units per unit area respectively)
Line 232. “pcs. on the mind unit sq.,” – corrected (units per unit area respectively)
Line 234. “units. on the mind unit sq.,” – corrected (units per unit area respectively)
Line 235. Figure 4. Check and change colors in numbers placed on photomicrograph images - corrected
Line 246. “markers. on the mind unit sq” – corrected (units per unit area respectively)
Line 248. “pcs. on the mind unit sq.,” – corrected (units per unit area respectively)
Lines 253 and 254. “pcs. on the mind unit sq.,” – corrected (units per unit area respectively)
Line 256. “on the mind unit sq.” – corrected (units per unit area respectively)
Lines 262-265. Rephrase sentence “The search for histochemical studies of markers in the norm and pathologies of organ systems is an urgent task of modern laboratory medicine” – corrected (One of the urgent tasks of laboratory medicine is the study of immunohistochemical markers for pathological conditions of various genesis.)
Line 332. Conclusions should be rephrased. Most of the paragraphs in lines 335-348 are a repetition of the findings in the spleen immunohistochemical study described in the results section.- corrected (Our research has shown that the immunophysiological state of the body changes the physical work and psycho-emotional stress of animals. Thus, the immunoregulatory blood index of dogs after intensive exercise significantly decreased from 2.1 ± 0.1 to 1.7 ± 0.13. This change is mainly due to a decrease in T-helpers by 13.4% and a slight change in the number of T-suppressors (+2.2%). It should be noted that the recovery of hematological indicators did not fully occur within 24 hours. We also studied the peculiarities of the location of lymphocytes with CD4+ and CD8+ markers in the pulp of the spleen. They were located almost all in the white pulp, in the red pulp their share was only 3.4% and 1.9%, respectively, of the total number of populations. Certain features of their localization were established: lymphocytes with both CD4+ and CD8+ markers in the white pulp of the spleen were mainly concentrated in lymphoid nodules (60.7% and 68.8%); in the lymphoid nodules themselves, the populations of these lymphocytes were located unevenly; in the periarterial lymphoid sheaths, the white pulp was found in significant quantities (36.3% and 29.2%, respectively). It was calculated that the immunoregulatory index of the spleen is equal to 1.9 ± 0.11.).
Thank you very much for the report
All comments were taken into account and corrected
Reviewer 2 Report
Comments and Suggestions for Authors
The topic is very relevant, the authors determining some immunoregulatory index as the ratio of CD4+ 32 cells to CD8+ cells in healthy adult dogs before and after intensive exercices
The methodology is modern and complex, using histology, immunohistochemistry and statistical analysis. Some details about Immunohistochemistry staining are needed.
The results have revealed that, the immunoregulatory blood index of dogs decreased after intensive exercise. These results may open new research directions in immunoregulatory index in stressful conditions.
The conclusions are consistent with the evidence and arguments presented.
The references are very relevant, including also some relevant author’s previous experience in the field.
I suggest some corrections
1. 2. Paragraphs from lines 66-67 IRI increases and thus affects the risk of autoimmune damage to cellular structures – be more specific- in which conditions?
3. Line 71 stenosis- of what?
4. Paragraphs from lines 71-72 are repeated on lines 91-92
Comments on the Quality of English LanguageThe manuscript should be checked by a native English speaker
Author Response
Thank you very much for the reviewer's report
1 2.Paragraphs from lines 66-67 IRI increases and thus affects the risk of autoimmune damage to cellular structures – be more specific- in which conditions? - corrected (An increase in IRI above >2.5 affects the risk of autoimmune damage to cellular structures and corresponds to hyperactivity of the immune response in patients with type 2 diabetes and non-alcoholic fatty liver disease).
- Line 71 stenosis- of what? Corrected
- Paragraphs from lines 71-72 are repeated on lines 91-92 - Corrected
Thank you very much for the report
All comments were taken into account and corrected
Reviewer 3 Report
Comments and Suggestions for Authors
This is an important and well-written paper. The charts and Figures are clear and understandable.
Suggested revisions to the Abstract are shown below.
Type of the Paper (Article, Review, Communication, etc.) 1 Immunophysiological state of dogs according to the immuno- 2 regulatory index in blood and spleen 3
Oksana Dunaievska1
, Ihor Sokulskyi1
, Mykola Radzykhovskii2
, Bogdan Gutyj3*
, Olga Dyshkant2![]()
, 4
Zoriana Khomenko1
, Viktor Brygadyrenko4
5
1 Polissya National University, Stary Boulevard Str., 7, Zhytomyr, 10002, Ukraine 6
2 National University of Life and Environmental Sciences of Ukraine, Heroiv Oborony Str., 15, Kyiv, 03041, 7
Ukraine 8
3 Stepan Gzhytskyi National University of Veterinary Medicine and Biotechnologies Lviv, Pekarska Str., 50, 9
Lviv, 79010, Ukraine 10
4 Oles Honchar Dnipro National University, Gagarin av., 72, Dnipro, 49010, Ukraine 11
* Correspondence: bvh@ukr.net 12
Simple Summary: Immunodiagnostics is essential for studying possible changes in the immune 13 system, which, together with the nervous and endocrine systems, constitutes the only necessary 14 regulatory system of the body. Among such studies, the state of the cellular link of immunity is 15 critical. These studies play an essential role in preventing diseases and increasing the effectiveness 16 of diagnostics. The immunoregulatory index, which allows assessing the immunophysiological sta- 17 tus of the direction of adaptive processes and the risk of autoimmune damage to cellular structures, 18 is of leading importance in assessing the state of the immune system. The immunoregulatory index 19 is sensitive to various effects and diseases of the body: liver disease, obesity, mononucleosis, and 20 radionuclide contamination of the territory. Scientific studies demonstrate that this index depends 21 on the breed characteristics of animals, has seasonal fluctuations, and is proposed for determination 22 in the selection process for the genetic improvement of already existing breeds. In current condi- 23 tions, the use of service dogs to search for explosives, people under rubble, and narcotic substances 24 is increasing. It is essential to observe the state of health of animals because the quality of their 25 official duties will depend on it. In this study, the immunoregulatory index was evaluated in dogs, 26 which allows you to objectively assess the immune status of their body and prevent the develop- 27 ment of a secondary immunodeficiency state. 28
Abstract: In this study, the immunological characteristics of the dog's body were established, al- 29 lowing a quick reaction to any changes in the immune status and the development of an immunodeficiency- 30
ciency state. The immunoregulatory blood index was determined to indicate the ratio of T-helpers 31 and T-suppressors. The immunoregulatory index of the spleen was determined as the ratio of CD4+ 32 cells to CD8+ cells in the field of view of a microscope (eyepiece 10, objective 40) after obtaining 33 histological preparations according to generally accepted methods. It was found that the number of 34
T-helpers decreased by 0.13 1012/L, while the number of T-suppressors increased non-significantly 35 by 0.01 1012/L after intensive exercise during tasks after extensive exercise, when . After intensive exercise, the immunoregulatory 36 index was significantly reduced to 1.7. Lymphocytes with markers CD4+ and CD8+ were located al- 37 most all in the white pulp; in the red pulp, they were found alone, and their share was 3.4% and 38 1.9%, respectively. Lymphocytes with CD4+ markers in the white pulp of the spleen were mainly 39 concentrated in lymphoid nodules (60.7%), of which 20.1% were focused on the marginal zone, 40 with slightly less in the light center (19.4%) and the periarterial zone (18.1%). Lymphocytes with CD8+ 41 markers in the white pulp of the spleen were also mainly concentrated in lymphoid nodules, but 42 their number was 8.1% higher (68.8%). The immunoregulatory index of the spleen is 1.9. These 43 findings emphasize the need to assess the immunoregulatory index in service dogs to prevent the 44 development of secondary immunodeficiency and their proper performance of
official duties. 45
Author Response
Thank you very much for the report
All comments were taken into account and corrected
Round 2
Reviewer 2 Report
Comments and Suggestions for Authors
The authors have made all the corrections suggested. I propose the acceptance of the article in actual revised form.